# ONLINE HYPER-PARAMETER OPTIMIZATION

## ABSTRACT

We propose an efficient online hyperparameter optimization method which uses a joint dynamical system to evaluate the gradient with respect to the hyperparameters. While similar methods are usually limited to hyperparameters with a smooth impact on the model, we show how to apply it to the probability of dropout in neural networks. Finally, we show its effectiveness on two distinct tasks.

## 1 INTRODUCTION

With the growing size and complexity of both datasets and models, training times keep increasing and it is not uncommon to train a model for several days or weeks. This effect is compounded by the number of hyperparameters a practitioner has to search through. Even though search through hyperparameter space has improved beyond grid search, this task is still often computationally intensive, mainly because these techniques are offline, in that they need to perform a full learning before trying a new value. Recently, several authors proposed online hyperparameter optimization techniques, where the hyperparameters are tuned alongside the parameters of the model themselves by running short runs of training and updating the hyperparameter after each such run. By casting the joint learning of parameters and hyperparameters as a dynamical system, we show that these approaches are unstable and need to be stopped using an external process, like early stopping, to achieve a good performance. We then modify these techniques such that the joint optimization procedure is stable as well as efficient by changing the hyperparameter at every time step. Further, while existing techniques are limited in the type of hyperparameters they can optimize, we extend the process to dropout probability optimization, a popular regularization technique in deep learning.

## 2 RELATED WORK

Historically, optimizing hyperparameters was done by selecting a few values for each hyperparameter, computing the Cartesian product of all these values, then by running a full training for each set of values. Bergstra & Bengio (2012) showed that performing a random search rather than a grid search was vastly more efficient, in particular by avoiding spending too much time training models with one of the hyperparameters set to a poor value. This technique was later refined in Bousquet et al. (2017) by using quasi random search. However, the parameters and their ranges have to be selected in advance, and potentially many trainings have to be performed to find good parameters. To remedy this issue, Snoek et al. (2012) used Gaussian processes to model the validation error as a function of the hyperparameters. Each training further refines this function to minimize the number of sets of hyperparameters to try.

All these methods are "black-box" methods, in that they assume no knowledge about the internal training process to optimize hyperparameters. In particular, they are gradient-free since they do not have access to the gradient of the validation loss with respect to the hyperparameters.

To solve this issue, Maclaurin et al. (2015) and Pedregosa (2016) explicitly used the parameter learning process to obtain such a gradient. These techniques, however, still need to complete a full optimization between each update of the hyperparameters. This can be an issue for very long trainings as poor choices of hyperparameters are not discarded right away, leading to unnecessary computations.

The work most closely related to ours is that of Franceschi et al. (2017) where the hyperparameters are changed after a certain number of parameter updates. However, not only must that number of

updates be chosen manually, the proposed algorithm is not stable and moves away from the optimum after some time. While this issue can be solved using other techniques, e.g., early stopping, we propose a stable algorithm by casting the joint learning of parameters and hyperparameters as a dynamical system. We also show how convergence can be obtained by changing the hyperparameters after each parameter update, thus further simplifying the algorithm. We then propose modifications to improve the speed and robustness of the optimization. Finally, we demonstrate the performance of our method on several problems.

## 3 HYPERPARAMETER OPTIMIZATION

The goal of learning is to find parameters which minimize the true expected risk. As we do not have access to that risk, we rely instead on the minimization of the empirical risk obtained using a training set. However, it is well-known that this can lead to overfitting, which can be prevented by regularization. A common method to choose which and how much regularization to use is to hold out part of the training set and to find which regularization yields the best performance on that held-out, or validation, set. We denote by *parameters* the parameters of the function being learnt and by *hyperparameters* the parameters of the regularization being used.

Hyperparameter optimization looks for the hyperparameters $\lambda$ such that the minimization of the regularized training loss over model parameters $\theta$ leads to the best performance on the validation set. Using this nomenclature, the best hyperparameters are selected according to:

$$\lambda^\dagger = \arg\min_\lambda L_V(\theta^*(\lambda)) \quad \text{with} \quad \theta^*(\lambda) = \arg\min_\theta L_T(\theta, \lambda) \tag{1}$$

where $L_V$ is the unregularized validation loss and $L_T$ the regularized training loss. It is important to note that this work focuses exclusively on regularization hyperparameters. In particular, we do not attempt to optimize optimization parameters such as the learning rate.

Eq. (1) shows that, to determine an optimization strategy for $\lambda$, one may compute the gradient of $L_V(\theta^*(\lambda))$ with respect to $\lambda$, which we call the *hypergradient*, and perform gradient descent.

### 3.1 FULL OPTIMIZATION

By the chain rule, we have

$$\frac{\partial L_V(\theta^*(\lambda))}{\partial \lambda} = g_V(\theta^*(\lambda)) \frac{\partial \theta^*(\lambda)}{\partial \lambda} \ ,$$

where, to simplify notations, we denoted by $g_V$ the gradient of $L_V$ with respect to $\theta$, i.e. $g_V := \frac{\partial L_V}{\partial \theta}$. Similarly, we denote $g_T := \frac{\partial L_T}{\partial \theta}$ the gradient of the regularized training loss with respect to $\theta$.

Since, by definition of the optimum, we have $g_T(\theta^*(\lambda), \lambda) = 0$, we can use the implicit function theorem to get:

$$\frac{\partial \theta^*}{\partial \lambda}(\lambda) = -\frac{\partial g_T}{\partial \theta}(\theta^*, \lambda)^{-1} \frac{\partial g_T}{\partial \lambda}(\theta^*, \lambda) \,. \tag{2}$$

Several algorithms propose to compute the hypergradient exactly Maclaurin et al. (2015) compute this derivative by backpropagating through the whole training procedure. Unfortunately this is very costly both in memory footprint and in wall time as several training procedures need to be serialized, hence is not easily scalable to large models. Pedregosa (2016) computes an approximate derivative when the model parameters are close to the optimal ones. In both cases, one needs to perform a full, or almost full, optimization to compute the gradient, leading to expensive updates.

We shall now see how we can compute approximate updates using far fewer optimization steps.

### 3.2 ALTERNATING OPTIMIZATION

The core idea is that the convergence of iterates $\theta_t$ to $\theta^*$ should allow us to use these iterates to update $\lambda$ rather than wait until convergence. In doing so, we could optimization the hyperparam-

eters simultaneously with the optimization of the model parameters. This idea has been explored by Franceschi et al. (2017) who proposed to optimize the validation error obtained when running exactly $K$ steps of gradient descent with fixed hyperparameters, i.e.

$$\min_{\lambda} \quad L_V(\theta_K(\lambda, \theta_0)) \tag{3}$$

$$\text{subject to:} \quad \theta_{t+1} = \theta_t - \eta g_T(\theta_t, \lambda) \tag{4}$$

where the constraint corresponds to the updates of $\theta$ using gradient descent with a learning rate $\eta$ and where $\theta_t$ $(t > 0)$ implicitly depends on $\lambda$.

The $K$-iterate hypergradient is then given by:

$$\frac{\partial L_V(\theta_K(\lambda, \theta_0))}{\partial \lambda} = g_V(\theta_K)\frac{\partial \theta_K(\lambda, \theta_0)}{\partial \lambda} . \tag{5}$$

Computing $\frac{\partial \theta_K}{\partial \lambda}$ can be done recursively by differentiating the gradient update recurrence in Eq. (4) with respect to $\lambda$:

$$\frac{\partial \theta_{t+1}}{\partial \lambda} = \frac{\partial \theta_t}{\partial \lambda} - \eta \left( \frac{\partial g_T}{\partial \theta}(\theta_t, \cdot)\frac{\partial \theta_t}{\partial \lambda} + \frac{\partial g_T}{\partial \lambda}(\theta_t, \cdot) \right)$$

Defining $y_t = \frac{\partial \theta_t}{\partial \lambda}$, we have two dynamical systems:

$$\theta_{t+1} = \theta_t - \eta g_T(\theta_t, \lambda) \tag{6}$$

$$y_{t+1} = y_t - \eta \left( \frac{\partial g_T}{\partial \theta}(\theta_t, \lambda)y_t + \frac{\partial g_T}{\partial \lambda}(\theta_t, \lambda) \right) . \tag{7}$$

starting from $\theta_0 = 0$, $y_0 = 0$. It is important to emphasize that, although we care about the convergence of the second system to compute the hypergradient, its trajectory is completely determined by that of $\theta_t$ and thus by the first system. In other words, the value of $y_t$ does not affect the optimization process over $\theta$.

Even though the system defined in Eq. (7) converges to the right solution, it can do so very slowly. For instance, assume that we are at hyperparameter $\lambda_0$ and that $\theta_0$ is initialized to the optimal value, i.e. $\theta_0 = \theta^*(\lambda_0)$. In that case, the system defined in Eq. (6) is already at convergence and $\theta_t = \theta^*(\lambda_0)$ for all $t$. The second system, however, will take some time to converge to the final value $\frac{\partial \theta^*(\lambda, \theta_0)}{\partial \lambda}$. Fig. A in the Appendix depicts the issue.

We now study the behavior of $y_t = \frac{\partial \theta_t}{\partial \lambda}(\lambda_0)$ whose recurrence is:

$$y_{t+1} = y_t - \eta \left( Ay_t + B \right)$$

$$\text{with} \begin{cases} A = \frac{\partial g_T}{\partial \theta}(\theta^*(\lambda_0), \lambda_0) = \frac{\partial^2 L_T}{\partial \theta \partial \theta^T}(\theta^*(\lambda_0), \lambda_0) \\ B = \frac{\partial g_T}{\partial \lambda}(\theta^*(\lambda_0), \lambda_0) . \end{cases}$$

The fixed point of this recurrence is $y^* = -A^{-1}B$ which is equal to the true hypergradient $\frac{\partial \theta^*}{\partial \lambda}(\lambda_0)$ according to Eq. (2). The convergence rate of $y_t$ depends on the spectrum of $I - \eta A$. If $\eta$ is too small, convergence will be slow and using a fixed number of steps $K$ can lead to a poor estimation of $\frac{\partial \theta^*}{\partial \lambda}$.

This poor estimation is mitigated by the fact that, if $\eta$ is small and $y_0 = 0$, $\|y_K\|$ will be in $O(K\eta)$ and thus the steps taken in hyperparameter space will also be small. We thus believe the overall effect of a small $\eta$ on the hyperparameter optimization will be limited to a smaller convergence.

We described how to perform one hyperparameter update using gradient descent with the hypergradient $\frac{\partial L_V(\theta_K)}{\partial \lambda}$. Franceschi et al. (2017) repeat this process in an outer loop, each time setting $\theta_0$ to the previous $\theta_K$ and initializing $y_0$ to 0. This reinitialization of $y$ at the beginning of each inner loop prevents the optimization from capturing any long term dependency of $\lambda$ on $\theta$ and $y_t$ from converging to the true hypergradient.

We now propose another formulation which maintains a growing history of the dependency of $\lambda$ on $\theta$, yielding increased stability.

### 3.3 First order hyperparameter optimization

Using the method of Franceschi et al. (2017) with $K = 1$ updates the hyperparameters at every gradient step, as proposed by Luketina et al. (2016). They compute $y(\lambda_t) = -\eta \frac{\partial g_T}{\partial \lambda}(\theta_t, \lambda_t)$, so that the hypergradient is estimated by

$$\frac{\widehat{\partial L_V(\theta_t)}}{\partial \lambda} = < g_V(\theta_t), y(\lambda_t) > = -\eta < g_V(\theta_t), \frac{\partial g_T}{\partial \lambda}(\theta_t, \lambda_t) > \tag{8}$$

Under this formulation, minimizing the validation loss over $\lambda$ is equivalent to maximizing $< g_V(\theta_t), g_T(\theta_t, \lambda_t) >$ using a specific scaling $\eta$ for the learning rate.

Assuming we are at a $\theta$ lying on the manifold $\{\theta^*(\lambda) : \lambda\}$, then the hypergradient defined by Eq. (8) is proportional to

$$\frac{\widehat{\partial L_V}}{\partial \lambda} \propto -g_V(\theta^*(\lambda)) \frac{\partial g_T(\theta, \lambda)}{\partial \lambda}\bigg|_{\theta=\theta^*},$$

which is in general not equal to the true hypergradient

$$\frac{\partial L_V(\theta^*)}{\partial \lambda} = g_V(\theta^*)\frac{\partial \theta^*}{\partial \lambda} = -g_V(\theta^*)\frac{\partial g_T}{\partial \theta}(\theta^*, \lambda)^{-1}\frac{\partial g_T}{\partial \lambda}(\theta^*, \lambda).$$

The two hypergradients are only proportional when the Hessian $\frac{\partial g_T}{\partial \theta}$ is a multiple of the identity. This suggests these first order methods can fail to converge to a local optimum. However, their simplicity makes them good candidates for the early stages of the optimization.

### 3.4 Online optimization with moving estimates

Instead of reinitializing $y_0(\lambda_t)$ to $0$ after every hyperparameter update, another possibility is to initialize $y_0(\lambda_t)$ to the last value $y_K(\lambda_{t-1})$ obtained using the previous value of $\lambda$. While this is beneficial when $\lambda_t$ is close to $\lambda_{t-1}$, issues might arise earlier in the optimization. Indeed, stopping the system before convergence could yield a value of $y_t$ much larger than $y_\infty$, overstimating the norm of the gradient and leading to a large change in $\lambda$. Although reinitializing $y_0$ to $0$ every time is crude, it favors smaller values of $y_t$ and thus smaller changes in $\lambda$, increasing stability.

To keep this stability while maintaining as much information about $y$ as possible, we propose to modify recurrence $y_t$ by constraining $y_t$ to lie within a ball:

$$y_t = P_{B(r)}\left(y_{t-1} - \eta\left(\frac{\partial g_T}{\partial \theta}(\theta_{t-1}, \lambda)y_{t-1} + \frac{\partial g_T}{\partial \lambda}(\theta_{t-1}, \lambda)\right)\right) \tag{9}$$

where $P_{B(r)}$ is the projection on the ball of radius $r$ and is formally defined by $P_{B(r)}(x) = \frac{r}{\max(\|x\|, r)}x$.

Every time the norm is clipped, this is equivalent to changing the stepsize for $\lambda$ but not the direction of the gradient. However, due to the dynamical nature of the system, it also affects future updates. As the learning rate decreases, so does the probability of clipping since:

$$\left\|y_{t-1} - \eta\left(\frac{\partial g_T}{\partial \theta}(\theta_{t-1}, \lambda)y_{t-1} + \frac{\partial g_T}{\partial \lambda}(\theta_{t-1}, \lambda)\right)\right\|^2 - \|y_{t-1}\|^2 = O(\eta).$$

$r$ is a hyperparameter which was chosen in the experiments so that clipping occurs almost at every step at the beginning of optimization, behaving like the method of Luketina et al. (2016). Our proposed method is summarized in Algorithm 1.

---

**Algorithm 1** Online hyperparameter optimization.

1: **procedure** HYPERPARAMETEROPTIMIZATION(num_steps, r, enable_projection)
2: $\quad(\theta_0, \lambda_0) \leftarrow$ Initial value of the (parameters, hyperparameters)
3: $\quad y_0 \leftarrow 0$
4: $\quad \alpha_0 \leftarrow 0$
5: $\quad$**for** $t <$ num_steps **do**
6: $\qquad y_{t+1} \leftarrow P_{B(r)} \left( y_t - \eta \left( \frac{\partial g_T}{\partial \theta}(\theta_t, \lambda_t) y_t + \frac{\partial g_T}{\partial \lambda}(\theta_t, \lambda_t) \right) \right)$
7: $\qquad \theta_{t+1} \leftarrow \theta_t - \eta g_T$
8: $\qquad g_P \leftarrow g_V$
9: $\qquad$**if** $t \geq$ warmup_time **then**
10: $\qquad\quad \lambda_{t+1} \leftarrow \lambda_t - c\eta g_P y_t \quad$ with: $\quad c$ constant scaling
11: $\qquad$**else**
12: $\qquad\quad \lambda_{t+1} \leftarrow \lambda_t$

---

## 3.5 REGULARIZATION SPECIFICS

We now describe in more details how we optimized two different regularizers: $\ell_2$ penalty, which has been optimized previously using hyperparameter optimization techniques, and dropout probability, for which, to the best of our knowledge, no existing techniques can be applied.

### 3.5.1 $\ell_2$ REGULARIZATION

The training loss, in case of $\ell_2$ regularization, is given by $L_T(\theta, \lambda) = L(\theta) + \frac{1}{2}\lambda \|\theta\|^2$.

In neural networks, we can differentiate two types of linear layers depending on whether $L(\theta)$ is sensitive to the norm of the weights or not. For example, the unregularized loss does not depend on the norm when a linear layer is followed by a normalization layer like batch norm: any change of the norm is compensated by the normalization layer. In those cases, $\ell_2$ regularization does not prevent overfitting as the norm can be decreased arbitrarily close to 0 without changing the function represented by the neural network. However, it has an impact on the dynamic of the training. van Laarhoven (2017) showed that for such a layer represented by weights $\theta$, the effective learning rate is $\eta_{\text{eff}} = \dfrac{\eta}{\|\theta\|^2}$. Since the gradient is orthogonal to the vector of weights Salimans & Kingma (2016), the norm of the weights after a gradient update is $\left\| \theta - \eta \left( \frac{\partial L}{\partial \theta} + \lambda\theta \right) \right\|^2 = (1 - \eta\lambda)^2 \|\theta\|^2 + \eta^2 \left\| \frac{\partial L}{\partial \theta} \right\|^2$ and keeps increasing if there is no $\ell_2$ regularization (i.e. $\lambda = 0$). The norm remains stable after a gradient update only when $\|\theta\|^2 = \frac{\eta}{2\lambda - \eta\lambda^2} \left\| \frac{\partial L}{\partial \theta} \right\|^2$ Assuming a learning rate small enough such that $\eta\lambda << 1$, we have $\|\theta\|^2 \approx \frac{\eta}{2\lambda} \left\| \frac{\partial L}{\partial \theta} \right\|^2$. In terms of effective learning rate, the norm remains stable when $\eta_{\text{eff}} = \frac{2\lambda}{\left\| \frac{\partial L}{\partial \theta} \right\|^2}$, i.e. when the effective learning rate does not depend on the initial learning rate. This short analysis shows that $\ell_2$ regularization can have a significant impact on the optimization without having a proper regularization effect. In this paper, we do not intend to address the problem of optimizing hyperparameters that have only an impact on the dynamic of the training and focus on the original intent of $\ell_2$ regularization as a way to prevent overfitting.

### 3.5.2 DROPOUT

Introduced in Hinton et al. (2012) and further studied in Srivastava et al. (2014), dropout is a way to regularize by preventing co-adaptation of output units of a neural network. Regularization is achieved by considering an ensemble of network architectures which differ only by their connections between the output units and the input units of the next layer, the weights being shared. Each output unit can be either kept with probability $p$ or dropped, meaning there is no connection to the next layer. The keep/drop decision can be represented by a vector mask $m$ which indicates for each output unit whether this one is kept or dropped. The probability to keep an output unit is often considered as an hyperparameter of the model, denoted as $\lambda$ in this section. The training loss can be

computed as an expectation over dropout masks $m$:

$$\bar{L}_T(\theta, \lambda) = E_{m \sim B(p=\lambda)}[L_T(\theta, m)]$$

where $B(p = \lambda)$ denotes the Bernoulli distribution. To compute the dependencies between the state $\theta$ and the hyperparameter $\lambda$ (see Eq. (6) and (7)), we need to have access to $\frac{\partial \bar{L}_T}{\partial \lambda}$. This cannot be formally computed, but can be approximated with finite differences:

$$\frac{\partial \bar{L}_T}{\partial \lambda} \approx \frac{1}{2\epsilon} \left( E_{m \sim B(p=\lambda+\epsilon)}[L_T(\theta, m)] - E_{m \sim B(p=\lambda-\epsilon)}[L_T(\theta, m)] \right)$$

In order to minimize the complexity, we just sample one dropout mask for $p = \lambda + \epsilon$ and one for $p = \lambda - \epsilon$ and compute the approximate derivative of the loss. The variance of the derivative computed this way can be quite large though. Instead, we use the fact that:

$$B(p = p_1) = B(p = p_2)B(p = \frac{p_1}{p_2}) \quad \text{for} \quad p_1 < p_2 \tag{10}$$

to sample a mask for $p_1 = \lambda - \epsilon$ which is not independent from the mask sampled using $p_2 = \lambda + \epsilon$.

## 4   EXPERIMENTS

We compare several methods to train the hyperparameters in an online way:

(a) *Unroll1-gTgV*, a first order method directly maximizing $g_T g_V$ (Section 3.3, no $\eta$),

(b) *UnrollK*, the version from Franceschi et al. (2017) (Section 3.2) which optimizes the hyperparameters over a fixed training window of size $K$,

(c) *ClipR*, described in Algorithm 1 with a clipping threshold equal to $R$.

Note that *Unroll1* is a special case of *UnrollK* which differs from *Unroll1-gTgV* by the factor $\eta$ (which can have an influence when $\eta$ is depends on $t$).

We evaluate these methods on models of increasing complexity, starting with a toy problem and ending on a typical deep learning setup. As a baseline, we use the typical one-shot hyperparameter optimization where the model is learnt $N$ times, once for each value of the hyperparameters, keeping the hyperparameters achieving the best validation loss.

While conducting the evaluation, we should be aware of possible overfitting on the validation set: information leaked from that set should be the same as with the typical one shot optimization algorithm. In particular, tested methods contain hyper-hyperparameters, like the clipping threshold in method *ClipR*. Since our original goal is to simplify hyperparameter optimization, we also test how sensitive to the particular values of these hyper-hyperparameters the final result is.

Finally, we evaluate the intrisic stability of the various online algorithms. To do so, we shall compare the performance of each online hyperparameter optimization method with and without using early stopping. A large gap between these two values indicates the best hyperparameter value is not a stable point for this particular method. Again, we use as baseline the same gap computed when the training is done with fixed hyperparameters.

While we mostly report final results, more detailed reports of all these experiments are available in the appendix.

### 4.1   QUADRATIC FUNCTION

We consider a strongly convex optimization problem where the training loss and the validation loss are given by $L_T(\theta, \lambda) = \frac{1}{2}(\theta - \bar{\theta})^T H(\theta - \bar{\theta}) + \frac{1}{2}\lambda\theta^T H\theta$ and $L_V(\theta) = \frac{1}{2}(\theta - \tilde{\theta})^T H(\theta - \tilde{\theta})$ where $\bar{\theta}, \tilde{\theta}$ are of dimension 20 and $H$ is a diagonal matrix. The optimal $\lambda$ can be computed analytically and is equal to $\lambda^\dagger = \frac{\bar{\theta}^T H(\bar{\theta} - \tilde{\theta})}{\bar{\theta}^T H\tilde{\theta}}$.

For the first 20K steps, the learning rate $\eta$ is set to $10^{-3}$ and the parameters at the end of this first phase are denoted by $(\theta_1, \lambda_1)$. $\eta$ is then set to $10^{-4}$ for the following 20K steps and the parameters

| Method | $\Delta L_V(\theta_1)$ | $|\lambda_1 - \lambda^\dagger|$ | $\Delta L_V(\theta_2)$ | $|\lambda_2 - \lambda^\dagger|$ |
|---|---|---|---|---|
| Unroll1 | +0.77% | 0.047 | +0.77% | 0.047 |
| Unroll5 | +0.56% | 0.039 | +0.72% | 0.045 |
| Unroll10 | +0.37% | 0.032 | +0.71% | 0.044 |
| Clip2 | +0.07% | 0.012 | +0.07% | 0.012 |
| Clip5 | +0.00% | 0.000 | +0.00% | 0.000 |

Table 1: Quadratic function: we see that longer rollouts lead to lower validation losses and more accurate values for the regularization parameter $\lambda$. Not only do clipping methods achieve better results, they are also more stable.

| Method | Cross-entropy loss ($\times 10^3$) | | Stability |
|---|---|---|---|
| | With early stopping | Without early stopping | |
| Fixed $\lambda$ | 71.6 | 73.3 | +2.4% |
| Unroll1-gTgV | 71.9 | 73.2 | +1.8% |
| Unroll1 | 73.0 | 73.6 | +0.8% |
| Unroll5 | 73.7 | 74.3 | +0.8% |
| Unroll10 | 73.6 | 74.8 | +1.6% |
| Clip5 | 71.9 | 73.0 | +1.5% |
| Clip10 | 71.4 | 72.7 | +1.8% |
| Clip20 | 70.8 | 71.8 | +1.4% |

Table 2: Performance on MNIST with a 4-hidden layer network (lower is better). Clipping leads to the best results, with or without early stopping. In that case, small unrolls are the most stable.

at the end of this phase are $(\theta_2, \lambda_2)$. The learning rate for the hyperparameters is always set to: $0.1\eta$. Constants $\bar{\theta}, \tilde{\theta}$ and $H$ are sampled so that $\lambda^\dagger$ lie in $[0.2, 0.4]$.

We repeat each training 50 times using different $\bar{\theta}, \tilde{\theta}, H$, and we evaluate the performance of each method using two metrics: (a) the average distance between $\lambda_i (i \in 1, 2)$ and $\lambda^\dagger$, (b) the increase in validation loss compared to optimal value $\theta^*(\lambda^\dagger)$.

Table 4.1 shows that, in line with the theoretical observations, methods *UnrollK* methods are unable to estimate the hypergradient when the learning rate decreases, leading to a slight increase of the loss. Additionally, while $\left\|\frac{\partial \theta^*}{\partial \lambda}\right\|$ is between 3.0 and 5.0, using 2.0 as a clipping threshold leads to only a slight increase of the validation loss while *Clip5* converges to the correct value.

## 4.2 IMAGE CLASSIFICATION

We consider here a feedforward network with 4 fully connected layers, of size 100 for the 3 hidden layers and of size 10 for the last layer. It is trained on MNIST using $\ell_2$ regularization. We also use a decaying schedule for the learning rate as it has been shown to possibly impact the online hyperparameter optimization algorithms. The dataset is split between a training set (75% of the samples) and a validation set (25% of the samples). Every online algorithm is run 6 times with a different initialization of the weight decay, equally spaced in the log domain between $10^{-6}$ and $10^{-1}$. The results are averaged over those 6 runs. The metric we optimize for is the cross-entropy on the validation dataset.

Table 4.2 shows that clipping outperforms the other methods, especially when early stopping is not used, displaying higher stability. We also note that *Clip20* performs better than a fixed $\lambda$ found through grid search, showing the efficiency of the method on more realistic problems.

| Method | Cross-entropy loss ($\times 10^3$) | | Stability |
| --- | --- | --- | --- |
| | With early stopping | Without early stopping | |
| Fixed $\lambda$: $\lambda = 0.35$ | 84.9 | 85.2 | +0.4% |
| Fixed $\lambda$: $\lambda = 0.40$ | 84.3 | 86.3 | +2.3% |
| Fixed $\lambda$: $\lambda = 0.45$ | 85.5 | 88.2 | +3.1% |
| Unroll1-gTgV | 85.6 | 95.8 | +11.9% |
| Unroll1 | 86.1 | 92.6 | +7.6% |
| Unroll5 | 84.3 | 89.1 | +5.6% |
| Unroll10 | 84.2 | 87.4 | +3.8% |
| Clip r=5.0 | 83.2 | 83.8 | +0.8% |
| Clip r=10.0 | 83.1 | 83.7 | +0.7% |
| Clip r=20.0 | 84.2 | 84.7 | +0.6% |
| Clip r=5.0, LR scale in $\{1.0, 0.1\}$ | 83.7 | 84.4 | +0.8% |

Table 3: Performances on PTB with an LSTM (lower is better).

### 4.3 LANGUAGE MODEL

Finally, we consider a language modeling task using the PTB dataset (Marcus et al. (1993)) and a typical LSTM-based network architecture[1]. In this architecture, the use of dropout is critical in order to prevent overfitting on the training dataset. Training is done using a decaying learning rate with a multiplicative decay of $0.95$ every 5K mini-batches. The hyperparameter learning rate is chosen as a constant scaling of the parameter learning rate.

As described previously, the performance metric (the perplexity on the validation dataset in this case) in Table 3 are given with and without early stopping so as to derive a measure of intrisic instability of the online algorithms. Combined with early stopping, all the hyperparameter optimization methods achieve good performance with slightly worse results for *Unroll1-gTgV* and *Unroll1*.

However, the algorithms are not equivalent in terms of stability. Unrolling methods are the most unstable when close to convergence as they drift to higher probabilities of keeping the weights (under-regularization). As explained in Section 3.3, method *Unroll1* attenuates this effect compared to *Unroll1-gTgV* by using a lower effective learning rate when this drift occurs. Contrary to what we observed on MNIST, longer rollouts seem here to increase the stability. On the other side, none of the flavors of the gradient clipping algorithm is subject to this instability: the instability metric is low and of the same order as the one derived using a fixed keep probability of $\lambda = 0.35$.

Last, online optimization of the hyperparameter does not increase overfitting on the validation dataset compared to a typical one shot algorithm: the minimum of the validation loss for those algorithms being close to the one obtained with one shot hyperparameter optimization.

## 5 CONCLUSION

Progress in optimization methods led to faster model training but the multiplication of hyperparameters means that one often must train many models to find the best one, an inefficient process putting most tasks out of reach for most. By casting hyperparameter optimization as a dynamical system, similar to standard parameter optimization, we derive an extension to the works of Luketina et al. (2016) and Franceschi et al. (2017) which exhibits higher stability and performance.

However, many questions remain. First, even though the method can be applied to any number of hyperparameters, we restricted our experiments to just one. There might be dynamics that are yet to be understood. Second, we did not apply this method to the hyperparameters of the optimizer, such as the step size or the decaying factor. As such hyperparameters affect the dynamics of the system, it is likely that the methods to optimize them differ from ours. Finally, our method involves other hyperparameters, such as the clipping factor. While our experiments point to the fact that their impact is limited, it would be best to get rid of them entirely.

---

[1]Medium configuration of the tensorflow model from https://goo.gl/zZahPt

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

# A  CONVERGENCE OF THE DERIVATIVE

Fig. 1 shows that, even if we start at the optimal value $\theta^*(\lambda)$ for the parameters, the dynamical system $\{y_t\}$ will take some time to converge to the true solution.

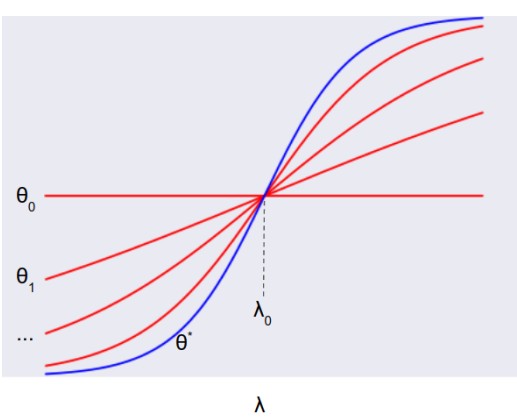

Figure 1: Illustration of the convergence of $\frac{\partial \theta_t}{\partial \lambda}(\lambda_0)$ when $\theta_0(\lambda)$ is a scalar initialized to $\theta^*(\lambda_0)$. $\theta_t$ has always the same value in $\lambda_0$ but its derivative converges to $\frac{\partial \theta_t}{\partial \lambda}(\lambda_0)$.

# B EXPERIMENTS: DYNAMIC OF THE TRAINING

We now show how the value of the hyperparameter and the training and validation loss vary during optimization. In particular, this will help determine when there are instabilities, i.e. when the best validation loss is not obtained at the end of the optimization.

## B.1 MNIST WITH $\ell_2$ REGULARIZATION

**Training using Unroll1-gTgV and Unroll1**

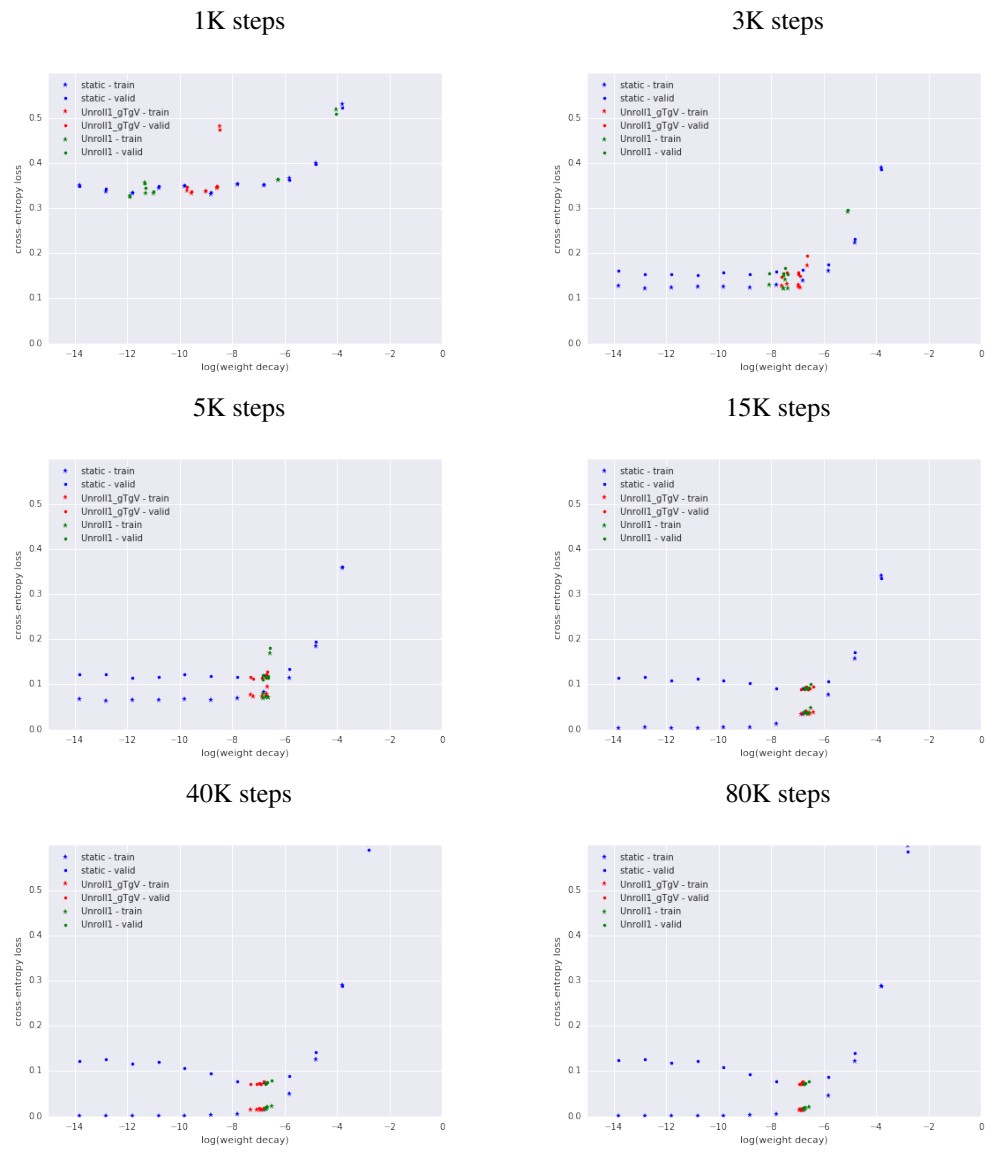

**Training using Unroll5 and Clip5**

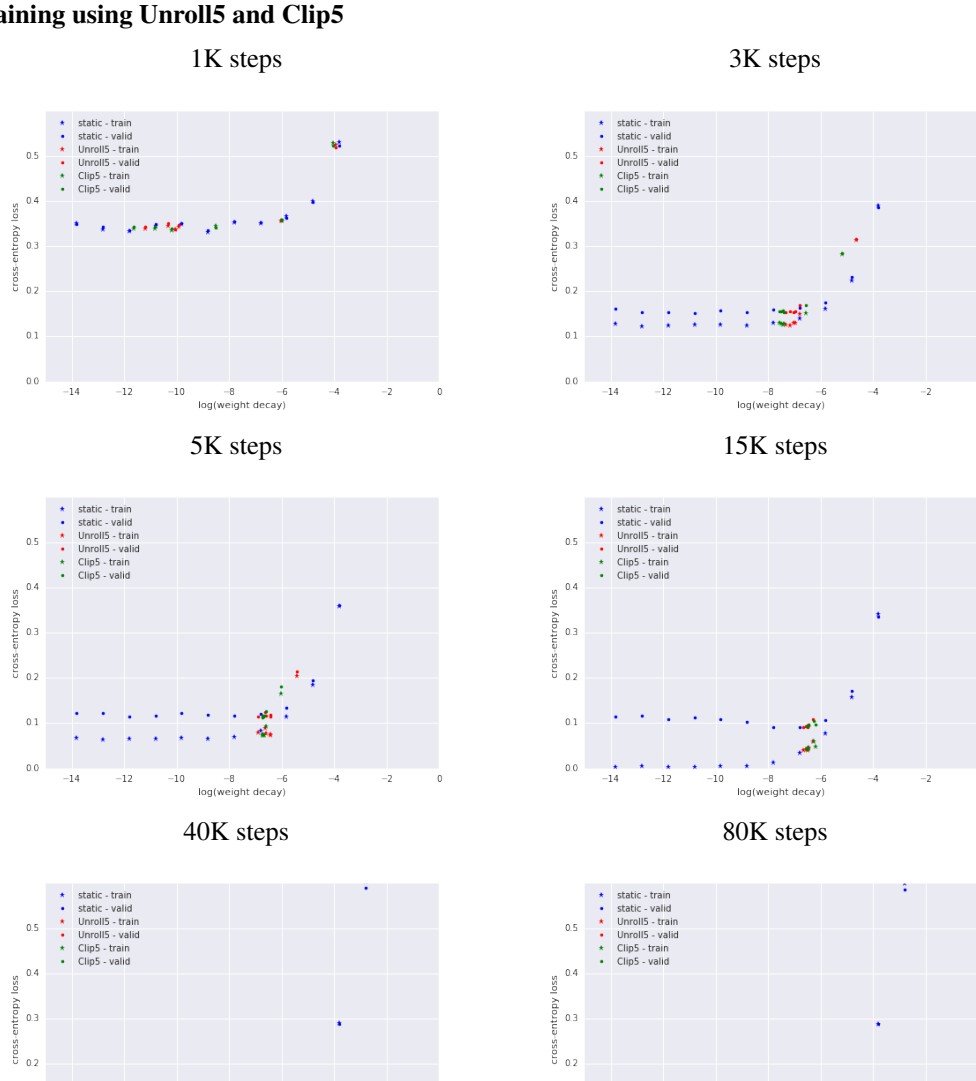

## B.2 PTB WITH DROPOUT

**Training using Unroll1-gTgV and Unroll1**

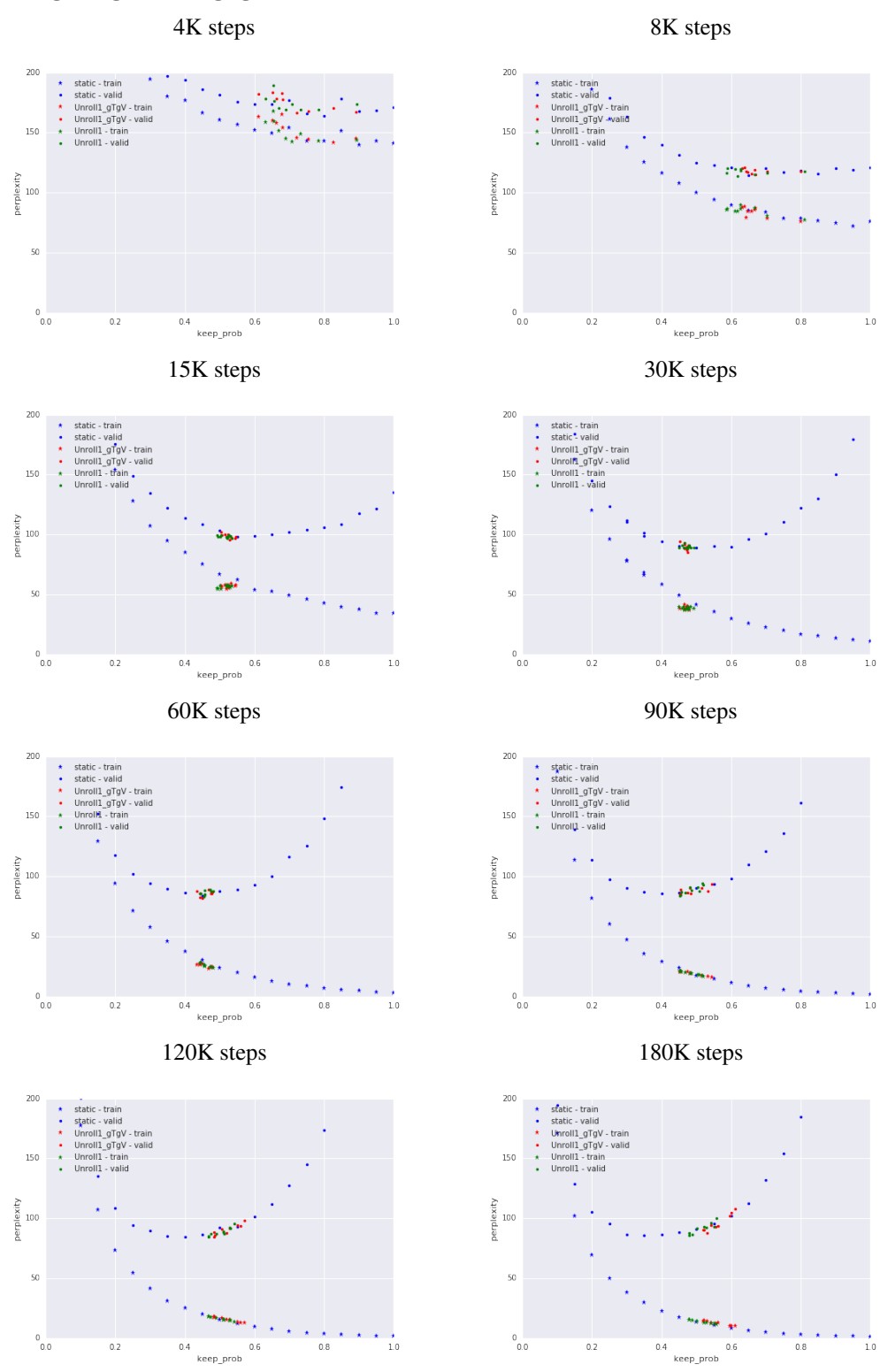

**Training using Unroll5 and Clip5**

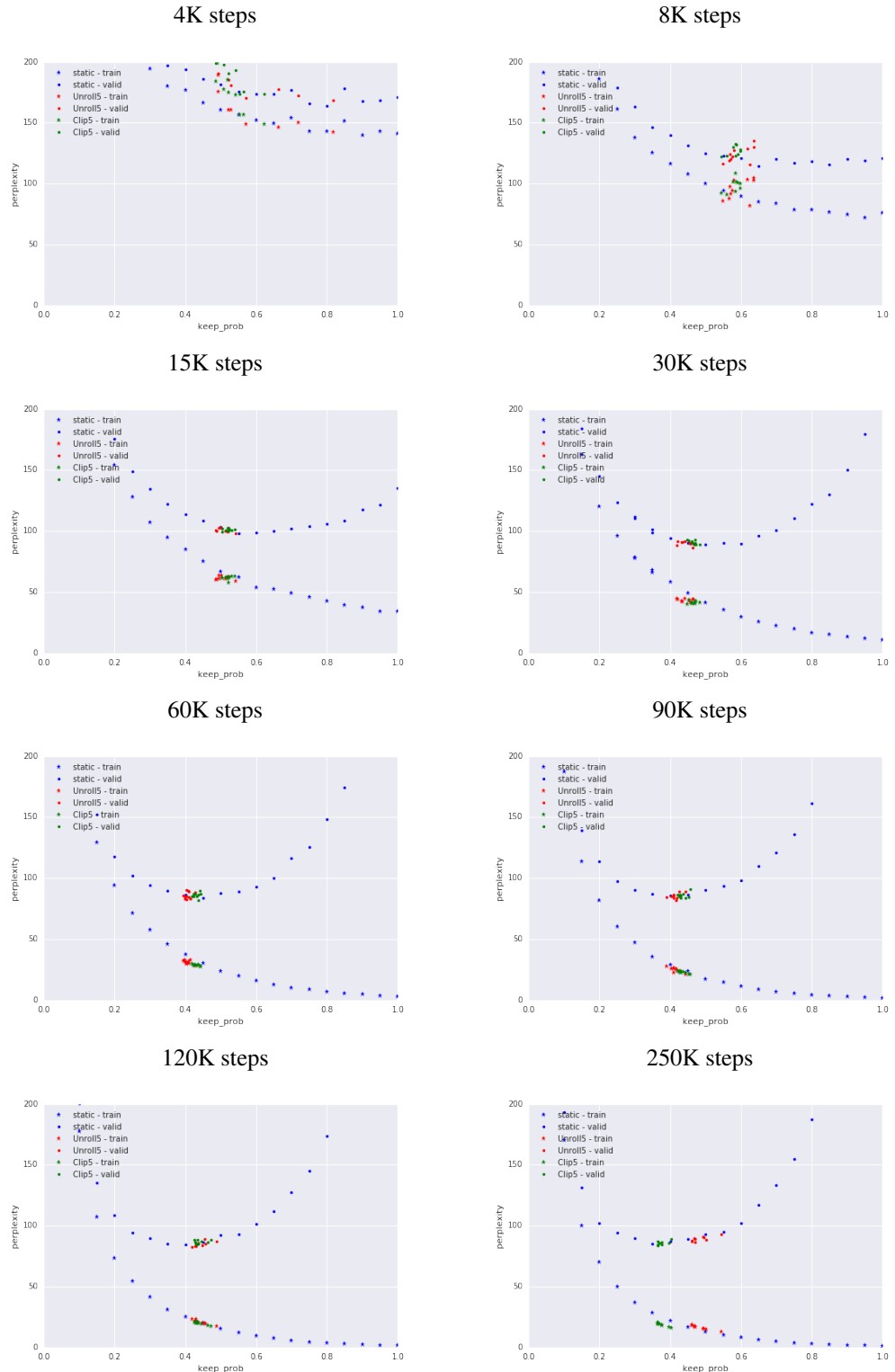

