# OpenReview forum: "Online Hyper-Parameter Optimization"
_ICLR.cc/2018/Conference — Reject_

### Official Review · AnonReviewer2 · 2017-11-27
**Online Hyper-Parameter Optimization**

**Rating:** 4
**Confidence:** 3

**Review:**

Summary of the paper
---------------------------
The paper addresses the issue of online optimization of hyper-parameters customary involved in deep architectures learning.  The covered framework is limited to regularization parameters. These hyper-parameters, noted $\lambda$, are updated along the training of model parameters $\theta$ by relying on the generalization performance (validation error). The paper proposes a dynamical system including the dynamical update of $\theta$ and the update of the gradient $y$, derivative of $\theta$ w.r.t. to the hyper-parameters. The main contribution of the paper is to propose a way to re-initialize $y$ at each update of $\lambda$ and a clipping procedure of $y$ in order to maintain the stability of the dynamical system. Experimental evaluations on synthetic or real datasets are conducted to show the effectiveness of the approach.

Comments
-------------
- The materials of the paper sometimes may be quite not easy to follow. Nevertheless the paper is quite well written.
- The main contributions of the paper can be seen as an incremental version of (Franceschi et al, 2017) based on the proposal in (Luketina et al., 2016). As such the impact of the contributions appears rather limited even though the experimental results show a better stability of the method compared to competitors.
- One motivation of the approach is to fix the slow convergence of the method in (Franceschi et al, 2017). The paper will gain in quality if a theoretical analysis of the speed-up brought by the proposed approach is discussed.
- The goal of the paper is to address automatically the learning of regularization parameters. Unfortunately, Algorithm 1 involves several other hyper-parameters (namely clipping factor $r$, constant $c$ or $\eta$) which choices are not clearly discussed. It turns that the paper trades a set of hyper-parameters for another one which tuning may be tedious. This fact weakens the scope of the online hyper-parameter optimization approach.
- It may be helpful to indicate the standard deviations of the experimental results.

---

### Official Review · AnonReviewer1 · 2017-12-02
**Interesting idea, weak execution**

**Rating:** 5
**Confidence:** 3

**Review:**


# Summary of paper
The paper proposes an algorithm for hyperparameter optimization that can be seen as an extension of Franceschi 2017 were some estimates are warm restarted to increase the stability of the method.

# Summary of review
I find the contribution to be incremental, and the validation weak. Furthermore, the paper discusses the algorithm using hand-waiving arguments and lacks the rigor that I would consider necessary on an optimization-based contribution. None of my comments are fatal, but together with the incremental contribution I'm inclined as of this revision towards marginal reject.

# Detailed comments

1. The distinction between parameters and hyperparameters (section 3) should be revised. First, the definition of parameters should not include the word parameters. Second, it is not clear what "parameters of the regularization" means. Typically, the regularization depends on both hyperparameters and parameters. The real distinction between parameters and parameters is how they are estimated: hyperparameters cannot be estimated from the same dataset as the parameters as this would lead to overfitting and so need to be estimated using a different criterion, but both are "begin learnt", just from different datasets.

2. In Section 3.1, credit for the approach of computing the hypergradient by backpropagating through the training procedure is attributed to Maclaurin 2015. This is not correct. This approach was first proposed in Domke 2012 and refined by Maclaurin 2015 (as correctly mentioned in Maclaurin 2015).

3. Some quantities are not correctly specified. I should not need to guess from the context or related literature what the quantities refer to. theta_K for example is undefined (although I could understand its meaning from the context) and sometimes used with arguments, sometimes without (i.e., both theta_K(lambda, theta_0) and theta_K are used).

4. The hypothesis are not correctly specified. Many of the results used require smoothness of the second derivative (e.g., the implicit function theorem) but these are nowhere stated.

5. The algorithm introduces too many hyper-hyperparameters, although the authors do acknowledge this. While I do believe that projecting into a compact domain is necessary (see Pedregosa 2016 assumption A3), the other parameters should ideally be relaxed or estimated from the evolution of the algorithm.

# Minor

missing . after "hypergradient exactly".

"we could optimization the hyperparam-" (typo)

References:
 Justin  Domke.    Generic  methods  for  optimization-based modeling.  In
International Conference on Artificial Intelligence and Statistics, 2012.

---

### Official Review · AnonReviewer4 · 2017-12-06
**Needs more work**

**Rating:** 4
**Confidence:** 3

**Review:**

Summary of paper:

This work proposes an extension to an existing method (Franceschi 2017) to optimize regularization hyperparameters. Their method claims increased stability in contrast to the existing one.

Summary of review:

This is an incremental change of an existing method. This is acceptable as long as the incremental change significantly improves results or the paper presents some convincing theoretical arguments. I did not find either to be the case. The theoretical arguments are interesting but lacking in rigor. The proposed method introduces hyper-hyperparameters which may be hard to tune. The experiments are small scale and it is unclear how much the method improves random grid search. For these reasons, I cannot recommend this paper for acceptance.

Comments:
1. Paper should cite Domke 2012 in related work section.
2. Should state and verify conditions for application of implicit function theorem on page 2.
3. Fix notation on page 3. Dot is used on the right hand side to indicate an argument but not left hand side for equation after "with respect to \lambda".
4. I would like to see more explanation for the figure in Appendix A. What specific optimization is being depicted? This figure could be moved into the paper's main body with some additional clarification.
5. I did not understand the paragraph beginning with "This poor estimation". Is this just a restatement of the previous paragraph, which concluded convergence will be slow if \eta is too small?
6. I do understand the notation used in equation (8) on page 4. Are <, > meant to denote less than/greater than or something else?
7. Discussion of weight decay on page 5 seems tangential to main point of the paper. Could be reduced to a sentence or two.
8. I would like to see some experimental verification that the proposed method significantly reduces the dropout gradient variance (page 6), if the authors claim that tuning dropout probabilities is an area they succeed where others don't.
9. Experiments are unconvincing. First, only one hyperparameter is being optimized and random search/grid search are sufficient for this. Second, it is unclear how close the proposed method is to finding the optimal regularization parameter \lambda. All one can conclude is that it performs slightly better than grid search with a small number of runs. I would have preferred to see an extensive grid search done to find the best possible \lambda, then seen how well the proposed method does compared to this.
10. I would have liked to see a plot of how the value of lambda changes throughout optimization. If one can initialize lambda arbitrarily and have this method find the optimal lambda, that is more impressive than a method that works simply because of a fortunate initialization.


Typos:
1. Optimization -> optimize (bottom of page 2)
2. Should be a period after sentence starting "Several algorithms" on page 2.
3. In algorithm box on page 5, enable_projection is never used. Seems like warmup_time should also be an input to the algorithm.

---

### Author Response · Authors · 2018-01-04
**Rebuttal**

We would like to thank the reviewers for their feedback.
We agree that the paper would benefit from some stronger theoretical justifications and/or more extensive experiments.
This will be part of some future work and we accept the decision of the reviewers to reject the paper.

Some answers to AnonReviewer4:
Thanks for your feedback.
#4: Figure in appendix A is a simplified view of the behavior of the 2 dynamical systems (given by Eq 6 and 7) when the first one (Eq 6) has reached convergence. The figure shows the convergence of the derivative (Eq (7)) in \lambda_0 can take some extra time after (6) has converged.
#5: The sentence mitigates what was previously said. When using a fixed number of steps, the estimated hyper-gradient could be far from the true hyper-gradient. However, the value of the hyper-parameter is not going to be significantly altered since the norm of the estimated gradient is a O(K \eta)  (Summary: the direction of the estimated hypergradient might be wrong but its norm is small).
#6: < x, y> in equation 8 denotes the inner product.
#9: We agree that the goal would be to apply this method on more than 1 hyper-parameter. We started with 1 hyper-parameter to gain a better understanding of the dynamic of the training.
#10: Appendix B.2 is showing some plots of the dropout parameter along the training with different initial values (from 0.1 to 0.9). As can be seen, for method "Clip5", all the points, regardless of the initialization of the hyper-parameter converge to the minimum of the validation loss.

Answer to AnonReviewer1:
Thanks for your comments. We will work to improve the paper to make it more rigorous.

Answer to AnonReviewer2:
Thanks for your comments.
We agree that the choice of the hyper-hyperparameter could be more extensively studied.
We have 2 hyper-hyperparameters that are specific to the estimation of the hyper-parameters: the clipping factor $r$ and the learning rate scale $c$. Table 3 tends to show that the algorithm is pretty robust w.r.t. the choice of the hyper-hyperparameters.

---

### Decision · Program_Chairs · 2018-01-29
**ICLR 2018 Conference Acceptance Decision**

**Decision:**

Reject

**Comment:**

This paper presents an update to the method of Franceschi 2017 to optimize regularization hyperparameters, to improve stability.  However, the theoretical story isn't so clear, and the results aren't much of an improvement.  Overall, the presentation and development of the idea needs work.